# Objectively Measured Sleep Duration and Health-Related Quality of Life in Older Adults with Metabolic Syndrome: A One-Year Longitudinal Analysis of the PREDIMED-Plus Cohort

**DOI:** 10.3390/nu16162631

**Published:** 2024-08-09

**Authors:** Alba Marcos-Delgado, Vicente Martín-Sánchez, Miguel Ángel Martínez-González, Dolores Corella, Jordi Salas-Salvadó, Helmut Schröder, Alfredo Martínez, Ángel M. Alonso-Gómez, Julia Wärnberg, Jesús Vioque, Dora Romaguera, José López-Miranda, Ramon Estruch, Francisco J. Tinahones, José M. Santos-Lozano, Jacqueline Álvarez-Pérez, Aurora Bueno-Cavanillas, Naomi Cano-Ibáñez, Carmen Amezcua-Prieto, Natalia Hernández-Segura, Josep A. Tur, Xavier Pintó, Miguel Delgado-Rodríguez, Pilar Matía-Martín, Josep Vidal, Clotilde Vázquez, Lidia Daimiel, Emili Ros, Estefanía Toledo, Tany E. Garcidueñas-Fimbres, Judith Viaplana, Eva M. Asensio, María D. Zomeño, Antonio Garcia-Rios, Alejandro Oncina-Cánovas, Francisco Javier Barón-López, Napoleón Pérez-Farinos, Carmen Sayon-Orea, Aina M. Galmés-Panadés, Rosa Casas, Lucas Tojal-Sierra, Ana M. Gómez-Pérez, Pilar Buil-Corsiales, Jesús F. García-Gavilán, Carolina Ortega-Azorín, Olga Castañer, Patricia J. Peña-Orihuela, Sandra González-Palacios, Nancy Babio, Montse Fitó, Javier Nieto

**Affiliations:** 1Faculty of Health Sciences, Department of Biomedical Sciences, Area of Preventive Medicine and Public Health, Universidad de León, 24007 León, Spain; vicente.martin@unileon.es (V.M.-S.); nhers@unileon.es (N.H.-S.); 2The Research Group in Gene-Environment and Health Interactions, Institute of Biomedicine (IBIOMED), Universidad de León, 24007 León, Spain; 3CIBER de Epidemiología y Salud Pública (CIBERESP), Instituto de Salud Carlos III, 28222 Madrid, Spain; dolores.corella@uv.es (D.C.); vioque@umh.es (J.V.); abueno@ugr.es (A.B.-C.); ncaiba@ugr.es (N.C.-I.); carmezcua@ugr.es (C.A.-P.); pep.tur@uib.es (J.A.T.); eva.m.asensio@uv.es (E.M.A.); carolina.ortega@uv.es (C.O.-A.); 4Centro de Investigación Biomédica en Red Fisiopatología de la Obesidad y la Nutrición (CIBEROBN), Institute of Health Carlos III, 28222 Madrid, Spain; mamartinez@unav.es (M.Á.M.-G.); jordi.salas@urv.cat (J.S.-S.); hschroder@imim.es (H.S.); jalfmtz@unav.es (A.M.); angelmago13@gmail.com (Á.M.A.-G.); jwarnberg@uma.es (J.W.); mariaadoracion.romaguera@ssib.es (D.R.); jlopezmir@gmail.com (J.L.-M.); restruch@clinic.cat (R.E.); fjtinahones@uma.es (F.J.T.); josemanuel.santos.lozano@gmail.com (J.M.S.-L.); jalvarez@proyinves.ulpgc.es (J.Á.-P.); xpinto@bellvitgehospital.cat (X.P.); cvazquezma@gmail.com (C.V.); lidia.daimiel@alimentacion.imdea.org (L.D.); eros@clinic.cat (E.R.); etoledo@unav.es (E.T.); tanyelizabeth.garciduenas@estudiants.urv.cat (T.E.G.-F.); mzomeno@imim.es (M.D.Z.); angarios2004@yahoo.es (A.G.-R.); msayon@alumni.unav.es (C.S.-O.); aina.galmes.panades@gmail.com (A.M.G.-P.); rcasasr@gmail.com (R.C.); lutojal@hotmail.com (L.T.-S.); anamgp86@gmail.com (A.M.G.-P.); jesusfrancisco.garcia@iispv.cat (J.F.G.-G.); patriciaj.penaorihuela@gmail.com (P.J.P.-O.); nancy.babio@urv.cat (N.B.); 5Department of Preventive Medicine and Public Health, Navarra Institute for Health Research (IdiSNA), University of Navarra, 31009 Pamplona, Spain; pilarbuilc@gmail.com; 6Department of Nutrition, Harvard T.H. Chan School of Public Health, Boston, MA 02124, USA; 7Department of Preventive Medicine, University of Valencia, 46008 Valencia, Spain; 8Unitat de Nutrició, Departament de Bioquímica i Biotecnologia, Universitat Rovira i Virgili, 43206 Reus, Spain; 9Institut d’Investigació Sanitària Pere Virgili (IISPV), 43204 Reus, Spain; 10Unit of Cardiovascular Risk and Nutrition, Institut Hospital del Mar de Investigaciones Médicas Municipal d’Investigació Médica (IMIM), 08003 Barcelona, Spain; ocastaner@imim.es (O.C.); mfito@imim.es (M.F.); 11Department of Nutrition, Food Sciences, and Physiology, Center for Nutrition Research, University of Navarra, 31009 Pamplona, Spain; 12Precision Nutrition and Cardiometabolic Health Program, IMDEA Food, CEI UAM + CSIC, 28222 Madrid, Spain; mdelgado@ujaen.es; 13Bioaraba Health Research Institute, Cardiovascular, Respiratory and Metabolic Area, Osakidetza Basque Health Service, Araba University Hospital, University of the Basque Country UPV/EHU, 01004 Vitoria-Gasteiz, Spain; 14EpiPHAAN Research Group, School of Health Sciences, Instituto de Investigación Biomédica de Málaga (IBIMA), University of Málaga, 29010 Málaga, Spain; aoncina@umh.es (A.O.-C.); baron@uma.es (F.J.B.-L.); napoleon.perez@uma.es (N.P.-F.); sandra.gonzalezp@umh.es (S.G.-P.); 15Instituto de Investigación Sanitaria y Biomédica de Alicante, Universidad Miguel Hernández (ISABIAL-UMH), 03202 Alicante, Spain; 16Health Research Institute of the Balearic Islands (IdISBa), 07012 Palma de Mallorca, Spain; 17Department of Internal Medicine, Maimonides Biomedical Research Institute of Cordoba (IMIBIC), Reina Sofia University Hospital, University of Cordoba, 30110 Cordoba, Spain; 18Department of Internal Medicine, Institut d’Investigacions Biomèdiques August Pi Sunyer (IDIBAPS), Hospital Clinic, Institut de Recerca en Nutrició y Seguretat Alimentaria (INSA-UB), University of Barcelona, 08001 Barcelona, Spain; 19Virgen de la Victoria Hospital, Department of Endocrinology, Instituto de Investigación Biomédica de Málaga (IBIMA), University of Málaga, 29010 Málaga, Spain; 20Research Unit, Department of Family Medicine, Distrito Sanitario Atención Primaria Sevilla, 41006 Sevilla, Spain; 21Research Institute of Biomedical and Health Sciences (IUIBS), University of Las Palmas de Gran Canaria, 35010 Las Palmas de Gran Canaria, Spain; 22Department of Preventive Medicine and Public Health, University of Granada, 18071 Granada, Spain; 23Instituto de Investigación Biosanitaria de Granada (ibs.GRANADA), 18071 Granada, Spain; 24Research Group on Community Nutrition & Oxidative Stress, University of Balearic Islands, 07012 Palma de Mallorca, Spain; 25Lipids and Vascular Risk Unit, Internal Medicine, Hospital Universitario de Bellvitge-IDIBELL, Hospitalet de Llobregat, 08901 Barcelona, Spain; 26Division of Preventive Medicine, Faculty of Medicine, University of Jaén, 23003 Jaén, Spain; 27Department of Endocrinology and Nutrition, Instituto de Investigación Sanitaria Hospital Clínico San Carlos (IdISSC), 28040 Madrid, Spain; mmatia@ucm.es; 28CIBER Diabetes y Enfermedades Metabólicas (CIBERDEM), Instituto de Salud Carlos III (ISCIII), 28222 Madrid, Spain; jovidal@clinic.cat (J.V.); viaplana@clinic.cat (J.V.); 29Department of Endocrinology, Institut d’Investigacions Biomédiques August Pi Sunyer (IDIBAPS), Hospital Clinic, University of Barcelona, 08001 Barcelona, Spain; 30Department of Endocrinology and Nutrition, Hospital Fundación Jimenez Díaz, Instituto de Investigaciones Biomédicas IISFJD, University Autonoma, 28049 Madrid, Spain; 31Nutritional Control of the Epigenome Group, Precision Nutrition and Obesity Program, IMDEA Food, CEI UAM + CSIC, 28222 Madrid, Spain; 32Departamento de Ciencias Farmacéuticas y de la Salud, Facultad de Farmacia, Universidad San Pablo-CEU, CEU Universities, Urbanización Montepríncipe, 28660 Boadilla del Monte, Spain; 33Lipid Clinic, Department of Endocrinology and Nutrition, Institut d’Investigacions Biomèdiques August Pi Sunyer (IDIBAPS), Hospital Clínic, 08001 Barcelona, Spain; 34Physical Activity and Sport Sciences Research Group (GICAFE), Institute for Educational Research and Innovation (IRIE), University of the Balearic Island, 07122 Palma, Spain; 35College of Public Health and Human Sciences, Oregon State University, Corvallis, OR 97330, USA; javier.nieto@oregonstate.edu

**Keywords:** sleep duration, daytime sleep duration, metabolic syndrome, health-related quality of life, quality of life, nap

## Abstract

The aim of our cross-sectional and longitudinal study is to assess the relationship between daytime and night-time sleep duration and health-related quality of life (HRQoL) in adults with metabolic syndrome after a 1-year healthy lifestyle intervention. Analysis of the data from 2119 Spanish adults aged 55–75 years from the PREDIMED-Plus study was performed. Sleep duration was assessed using a wrist-worn accelerometer. HRQoL was measured using the SF-36 questionnaire. Linear regression models adjusted for socioeconomic and lifestyle factors and morbidity were developed. In cross-sectional analyses, participants with extreme night-time sleep duration categories showed lower physical component summary scores in Models 1 and 2 [β-coefficient (95% confidence interval) <6 h vs. 7–9 h: −2, 3 (−3.8 to −0.8); *p* = 0.002. >9 h vs. 7–9 h: −1.1 (−2.0 to −0.3); *p* = 0.01]. Participants who sleep less than 7 h a night and take a nap are associated with higher mental component summary scores [β-coefficient (95% confidence interval) 6.3 (1.3 to 11.3); *p* = 0.01]. No differences between night-time sleep categories and 12-month changes in HRQoL were observed. In conclusion, in cross-sectional analyses, extremes in nocturnal sleep duration are related to lower physical component summary scores and napping is associated with higher mental component summary scores in older adults who sleep less than 7 h a night.

## 1. Introduction

Sleep habits in today’s society have changed compared with those of our ancestors. In general, we sleep less, have less quality sleep and are more unlikely to nap [1]. The main causes for this are suggested to be artificial light, using new technologies, working hours and the pace of life [1,2,3,4].

For decades, sleep-related problems have been increasing exponentially [5,6]; between 10 and 30% of the population reports sleeping less than 6 h a day (depending on country) [7], approximately 40% relates to having had a restful night’s sleep [8,9] and the consumption of sleep medications has skyrocketed [10], turning into a public health challenge.

According to the Survey of Health Ageing and Retirement in Europe, 24.3% of older adults in Spain have sleep problems [11], 54.3% reported sleeping less than seven hours (male 56.2% and female 52.8%) [9] and 16% take a daily nap, a percentage that increases with flexible working hours or retirement [11].

Evidence shows that a sleep deficit is a risk factor for increased mortality and major chronic diseases like cardiovascular diseases [12], hypertension, obesity, type 2 diabetes [13,14] or metabolic syndrome (MetS) [15]. Furthermore, a strong association has been shown between sleep duration and mental illnesses such as anxiety or depression [16,17].

In recent years, the paradigm in sleep research has shifted. For most of the last few decades, research focused mostly on sleep pathologies such as sleep apnea, insomnia and narcolepsy. In recent years, however, there has been increasing attention on a broader and more holistic approach, namely on the role of sleep health in all aspects of life and health in general [1,18]. Health-related quality of life (HRQoL), a measure of individual well-being, has been shown to be negatively impacted by sleep deficiency, either in terms of quality or quantity. However, it is not entirely clear which HRQoL components are most affected. Furthermore, most published studies are cross-sectional, use self-reported measures of sleep duration and/or do not take into account the effect that daytime sleep (naps) may have on HRQoL [19,20,21,22].

There is no consensus in the scientific literature regarding the effects of naps on health outcomes. It seems that long naps, lasting more than 1 h, may be a symptom or consequence of a nocturnal sleep disorder and have been found to be associated with an increased risk of obesity, type 2 diabetes or MetS [23,24]. However, short naps between 15 and 30 min/d have been associated with an increase in cognitive function, greater memory retention, recovery from fatigue and increased alertness [25,26,27].

Consequently, they could be a key element in improving the population’s HRQoL.

Therefore, the aim of our cross-sectional and longitudinal study is to assess the effects of daytime and night-time sleep on HRQoL in older adults with MetS after a 1-year healthy lifestyle intervention.

## 2. Materials and Methods

### 2.1. Study Design and Participants

The PREDIMED-Plus study is a 6-year ongoing, multicenter, controlled, randomized intervention study with two parallel groups for the primary prevention of cardiovascular disease, involving 6874 people recruited in 23 Spanish centers. The study methods have been reported elsewhere [28] and are available on the PREDIMED-Plus website (http://www.predimedplus.com, accessed on 1 May 2024). Eligible participants were community-dwelling men (55–75 years old) and women (60–75 years old), with a body mass index (BMI) between ≥27 and <40 kg/m^2^ and who met at least 3 components of the MetS definition [29]. The intervention group received an intense intervention with an energy-restricted traditional Mediterranean diet (erMedDiet), physical activity (PA) promotion and motivational support to lose weight. The control group received general recommendations about the Mediterranean diet and healthy guidelines. Out of 6874 participants, data derived from accelerometry were available in a subsample of 2223 participants. According to protocol, accelerometers were provided to a subset of participants (50% of participants in the intensive intervention group and 20% of those in the control group) in order to quantify PA and sleep. One hundred and four participants were excluded owing to incomplete sleep or covariate data. The final sample size was 2119 participants at baseline and 1-year follow-up (Appendix A).

The trial was approved by the Institutional Review Board of all recruitment centers where the study was conducted, according to the ethical standards of the Declaration of Helsinki. The trial was retrospectively registered in the International Standard Randomized Controlled Trial registry (ISRCTN: http://www.isrctn.com/ISRCTN89898870, accessed on 1 May 2024). All participants provided written informed consent.

### 2.2. Principal Predictor Variable: Objectively Assessed Sleep by Accelerometry

The participants wore a wrist-worn accelerometer on their nondominant wrist for 8 consecutive 24 h days (GENEActiv, ActivInsights Ltd., Kimbolton, UK). The GENEActiv contains a triaxial accelerometer capturing accelerations in a range of  ±8 G’s and it was set at a sampling rate of 40 Hz. The monitor is totally waterproof, and they were asked to not remove it during water-based activities (i.e., showering, bathing or swimming). Raw data were downloaded at each study center using the GENEActiv PC software 1.2 (ActivInsights Ltd., Cambridgeshire, UK) as binary files (bin) and were uploaded to a common server at the study coordinating center at the University of Malaga. All raw data files were processed on an ongoing basis with the open-source R package GGIR v. 2.4–3 (https://cran.r-project.org, accessed on 1 May 2024) [30].

The sleep detection algorithm HDCZA (the Heuristic Algorithm looking at Distribution of Change in Z-Angle), available as part of the GGIR package [31], was used to identify the sleep period time window (SPT window, which refers to the time window starting at sleep onset and ending when the person wakes up after the last sleep episode of the night). Open-source software such as GGIR allows raw data to be processed in an identical manner regardless of monitors, and equivalent values for sleep estimates have been shown between the most common monitors used in epidemiologic studies [32]. Daytime napping was estimated as a period of sustained inactivity during the day, detected as the absence of change in arm angle greater than 5 degrees for at least 5 min [31].

Days when the accelerometer registered a valid night record were considered as valid, and only results from participants with at least two valid record days (and nights) were included in the analyses.

### 2.3. Outcome Variable: Health-Related Quality of Life (HRQoL)

The dependent variable was HRQoL at baseline and 1-year follow-up, measured using the Spanish version of the SF-36 questionnaire [33,34]. This questionnaire consisted of 36 items that assessed eight dimensions or scales: physical function (PF), physical role (RP), bodily pain (BP), general health (GH), vitality (VT), social function (SF), emotional role (RE) and mental health (MH). These dimensions were used to define two health component summaries: the physical component summary (PCS) and the mental component summary (MCS). Each item received a numerical score that was encoded, summed up and put on a scale from 0 to 100. The higher the score, the better quality of life in the analyzed field [35]. The reliability of the scale used to determine the values of the Spanish population of ≥60 years was higher than the proposed standard of Cronbach’s α, 0.7 [36], and has been previously used to measure HRQoL in older adults [37,38].

### 2.4. Covariates

At baseline, self-reported information was obtained by interview for sociodemographic variables: sex (men/women), age (years), marital status (married or living with a partner, divorced or widowed or single), labor status (active, retired, unemployed or household work) and educational level (≤primary, secondary or university). The lifestyle covariates of smoking status (current, former or never smoker), caffeine drinks/day (mg/d), alcohol drinks/day (g/d) and leisure time spent watching TV (h/wk) were also collected through the baseline questionnaire. The Mediterranean diet adherence was measured through a 17-item questionnaire by a trained interviewer [39].

Body mass index (kg/m^2^) was calculated from weight and height, measured under standardized conditions with light clothing and no shoes, using electronic scales and portable extendable stadiometers. Mean values of the two measurements were used for the analyses according to the PREDIMED-Plus protocol. Physical activity intensities were objectively estimated by accelerometry and calculated and classified using a previously proposed threshold for ENMO (Euclidean Norm Minus One) in the nondominant wrist: inactivity (<45 mg), light activity (45–99.9 mg) and moderate-to-vigorous activity (MVPA, >100 mg) [40].

PA was categorized, according to WHO recommendations, as ≥150 min/week (active/non active) [41].

Finally, the following physician-diagnosed diseases were self-reported: hypertension (yes/no), type 2 diabetes mellitus (yes/no), sedative treatment (yes/no), depression (yes/no), sleep apnea (yes/no) and chronic obstructive pulmonary disease (yes/no).

### 2.5. Statistical Analysis

#### 2.5.1. Cross-Sectional Analysis

First, the analysis of baseline characteristics was carried out in the entire sample according to four categories of night-time sleep duration, as defined in previous studies (<6, ≥6–<7, ≥7–<9 and ≥9 h/day (h/d)) [19,21]. Means (standard deviation, SD) or medians (interquartile range, IQ) were used for quantitative variables with normal or non-normal distribution, respectively, and absolute and relative frequencies (*n*, %) were used for qualitative variables. *p*-Values for differences between categories of night-time sleep duration were calculated using a chi-squared test or ANOVA for categorical and continuous variables, respectively. In cases of non-normally distributed continuous variables, we performed a Kruskal–Wallis test.

Second, linear regression models were developed with HRQoL after 1-year follow-up as the dependent variable and sleep duration as the main independent variable (using ≥7–<9 h of sleep as reference category). Model 1 was a linear model adjusted for age and sex; Model 2 was Model 1 plus marital status, labor status and educational level; Model 3 was Model 2 plus lifestyle factors (smoking status, caffeine drinks/day, alcohol drinks/day, leisure time spent watching TV, adherence to a Mediterranean diet, BMI, PA and daytime sleep duration); and Model 4 was Model 3 plus morbidity (hypertension, type 2 diabetes, sedative treatment, depression, sleep apnea and chronic obstructive pulmonary disease) and treatment assignment. The Benjamini–Hochberg procedure (BH) was used as a correction test.

Because, in most cases, the results from Models 1 and 2 were virtually identical, only the latter are shown in some tables. Stratified analyses by sex were also performed (Appendix A).

Third, linear regression models were used to study the associations between daytime sleep duration and HRQoL, stratified by night-time sleep duration. The categories for daytime sleep duration were <15, ≥15–<60 and ≥60 min/day (min/d). Analyses using 4 categories—splitting the middle category in two using 30 min/d as the cut-off—yield similar results. The category <15 min/d of daytime sleep was used as the reference. The categories for night-time sleep duration were <7, ≥7–<9 and ≥9 h/d. We could not maintain four sleep categories due to the small number of participants who slept <6 h/d at night and had <15 min/d of daytime sleep. Linear regression models were as described above, removing daytime sleep duration in Model 3. It was not possible to perform stratified analyses by sex due to insufficient samples in the categories of interest.

#### 2.5.2. Longitudinal Analysis

To examine whether night- and daytime sleep duration predicted changes in HRQoL, linear regression was carried out, where the dependent variable was differences in HRQoL between baseline and one year after the intervention and night/day sleep duration was the main independent variable.

Twelve-month changes in HRQoL were expressed as means (SD) and β-regression coefficients and used to assess the 1-year average changes in HRQoL associated with night- and daytime sleep duration at baseline. A positive β-coefficient means an improvement in HRQoL, while a negative coefficient means a worsening.

We defined “clinically significant” changes in HRQoL when there was at least a 5% change in the magnitude of the SF-36 score. Logistic regression models were carried out with this cut-off point to define the dependent variable. In these models, odds ratios (ORs) were calculated: an OR above 1 quantifies the likelihood of a clinically significant improvement in participants’ HRQoL, while an OR less than 1 indicates the risk of that change being for the worse.

The categories of night- and daytime sleep were as described above, as were the models and potential confounders for which the analyses were adjusted. Stratified analyses by sex, age, BMI and PA were also performed except for analyses according to daytime sleep categories due to insufficient sample size.

The analysis was performed with StataCorp (College Station, TX, USA) 2019, Stata Statistical Software: Release 16.

## 3. Results

### 3.1. Baseline Descriptive Characteristics

Baseline characteristics in the total sample and according to four categories of night-time sleep duration are shown in Table 1. The night-time sleep duration mean (SD) in the total sample (*n* = 2119) was 8.0 (1.3) h/d, the mean age (SD) was 65.0 (4.9) years and 47.4% of participants were female. Those participants sleeping ≥9 h were older and more likely to be women, retired, have a university degree and never have smoked. They were also more likely to have lower light and moderate–vigorous PA levels, higher prevalence of depression and higher sedative treatment use. The individuals sleeping <6 h showed higher caffeine and alcohol consumption, spent more time watching TV on the weekends and had higher daytime sleep duration. No significant differences in marital status, adherence to energy-restricted MedDiet, hypertension, type 2 diabetes, sleep apnea and chronic lung disease were found. Regarding the SF-36 score, lower scores are observed for extreme sleep durations, obtaining statistically significant differences in all scales except for RE.

### 3.2. Cross-Sectional Analysis

#### 3.2.1. Night-Time Sleep Duration

Compared with those in the ≥7–<9 h category, participants in categories outside the recommended range of sleep duration (<6 h and ≥9 h) showed lower PF, BP, VT, SF, GH and PCS scores in the multivariable-adjusted Models 1 and 2 (Table 2 and Figure 1). When confounding lifestyle variables were included in the model (Model 3), statistically significant association was lost in all dimensions except for BP and PCS, although there was still a decrease in scores in those individuals who sleep more than 9 h. In the fully adjusted model (Model 4), the association between sleep duration and HRQoL dimensions was lost.

Model 3 was a linear model adjusted for sociodemographic variables: age, sex, marital status (married or living with a partner, divorced or widowed or single), labor status (active, retirement and unemployed or household work), educational level (≤primary, secondary and university) and lifestyle factors, such as smoking status (current, former or never smoker), caffeine drinks/day (mg/d), alcohol drinks/day (g/d), leisure time spent watching TV (h/wk), adherence to a Mediterranean diet measured through a 17-item questionnaire (continuous), BMI (kg/m^2^) and MVPA recommendations (active/inactive). Model 4 was a linear model adjusted in the same way as Model 3 and for morbidities, such as hypertension (yes/no), type 2 diabetes mellitus (yes/no), sedative treatment (yes/no), depression (yes/no), sleep apnea (yes/no) and chronic obstructive pulmonary disease (yes/no), daytime sleep (min/day) and treatment assignment, stratified by daytime sleep duration.

In the analysis stratified by sex, the women who slept <6 h or ≥9 had a lower score in the PCS (Models 1, 2 and 3), but when adjusted for morbidity, only those women who sleep less than 6 h have their PCS affected. In men, only sleeping <6 h was associated with a worse PCS in Model 1 ( Appendix A).

#### 3.2.2. Daytime Sleep Duration

Those participants who sleep less than 7 h a night and take a nap greater than or equal to 15 min/d show an improvement in MCS, while for those who sleep 7 h or more at night, no statistically significant association between napping and HRQoL was found (Table 3 and Figure 2).

Model 4 was a linear model adjusted for age, sex, marital status (married or living with a partner, divorced or widowed or single), labor status (active, retirement and unemployed or household work), educational level (≤primary, secondary or university), lifestyle factors, such as smoking status (current, former or never smoker), caffeine drinks/day (mg/d), alcohol drinks/day (g/d), leisure time spent watching TV (h/wk), adherence to a Mediterranean diet assessed using a 17-item questionnaire (continuous), BMI (kg/m^2^), MVPA recommendations (active/inactive), and for morbidities, such as hypertension (yes/no), type 2 diabetes mellitus (yes/no), sedative treatment (yes/no), depression (yes/no), sleep apnea (yes/no), chronic obstructive pulmonary disease (yes/no) and treatment assignment, stratified by night-time sleep duration.

### 3.3. Longitudinal Analysis

#### 3.3.1. Night-Time Sleep Duration

No differences between night-time sleep duration categories and 12-month changes in HRQoL were observed (Appendix A). We also found no association in logistic regression models analyzing clinically significant changes in the HRQoL and night-time sleep categories, except for the BP dimension, where participants sleeping <6 h were 47% more likely to have a significant improvement in BP than those sleeping between 7 and 9 h/d (Appendix A).

#### 3.3.2. Daytime Sleep Duration

Table 4 and Figure 3 shows a longitudinal association between daytime sleep duration categories (stratified by night-time sleep duration) and 12-month clinically significant changes in HRQoL. Among individuals who slept between 7 and 9 h, a nap between 15 and 30 min/d was associated with a significant worsening of PCS with respect to those who sleep less than 15 min/d. The same occurs in those individuals who sleep >9 h/d and who nap ≥30 min/d compared to those who sleep <15 min/d. If we perform the analyses with three categories of napping (<15 min/d (ref), ≥15–<60 min/d and ≥60 min/d), as in the cross-sectional analyses, the association is lost for all models except for those individuals who sleep more than 9 h/d, for which napping (any duration) predicts a worsening of the physical spheres in the fully adjusted model (Appendix A).

## 4. Discussion

In our study, we found that HRQoL is higher among individuals who adhere to the US National Sleep Foundation’s recommendations [11,42] (between 7 and 8 h of sleep each night) than among those in extreme categories of sleep duration. This association, however, appears to be at least partially explained (or mediated) by lifestyle variables or, especially, by the presence of co-morbidities.

A correlation between daytime sleep duration (naps) and improvement in MCS in individuals who have a short night’s sleep (<7 h) is shown in the analyses. This is especially novel since, to our knowledge, no studies have been found that analyze this relationship with an objective measure of daytime sleep duration.

The finding of a U-shaped relation between sleep duration and HRQoL is consistent with previously published studies. In a study on the US population published in 2018 [19], the relationship between sleep duration and self-reported HRQoL was analyzed in individuals with the most prevalent chronic diseases (*n* = 277,757) compared to healthy individuals (*n* = 172,052). Researchers observed that participants who slept less than 6 h and more than 10 h had a higher risk of obtaining unfavorable scores in the four measured indicators of HRQoL, both in the group with chronic diseases and in the healthy group. However, unlike our observations, these results remained unchanged after adjusting for confounding variables of lifestyle.

In a study carried out on a cohort of elderly people living in the community in Spain [21], a cross-sectional and a three-year longitudinal analysis documented the relationship between sleep duration and SF-36 scores. A U-shaped relationship was observed in women only, after adjusting for potential confounding factors. In our sex-stratified analyses, adjusted for potential confounders, we found a decrease in PCS in women who sleep less than 6 h, while this association becomes weaker among women who sleep more than 9 h and the MCS when introducing confounding morbidity variables (*p* = 0.07). As in the other study, no association was found in men.

The results of the longitudinal analyses in that study agree with ours in that it does not seem that a clinically significant change in HRQoL is related to the duration of sleep at night. The inconclusive result in our study may stem from the fact that a one-year follow-up time between the two measurements might not be enough time to assess the effect of sleep on HRQoL. In the study by Faubel et al. [21], the sleep duration was self-reported and an overestimation of the hours of sleep may be influencing the results.

However, our results show an exception in the BP scale. There appears to be a weak protective effect of short sleep duration and a clinically significant improvement in bodily pain [OR 1.47 (95% CI 1.00–2.16); *p* = 0.05] in the four adjusted models. We have found no previous studies that explain this relationship. A possible explanation may be that individuals with bodily pain have a low tolerance to lying down for a long period of time, so sleeping fewer hours may improve their long-term bodily pain. However, some studies show an inverse association between short sleep duration and bodily pain [43,44].

On the other hand, not all previous research finds a U-shaped association between sleep duration and HRQoL. In a study carried out in a sample of patients with type 2 diabetes [14], where sleep duration, measured by actigraphy, and the HRQoL were evaluated, a linear relationship was observed. For every 60 min increase in sleep duration, PCS decreased its score by 1.3 units, but no association was found with MCS. Similarly, in a study conducted in young adults from the US using national survey data (NHANES 2005–2008), an association between sleep duration and quality of life was found. Young adults who slept less than 7 h were more likely to report low physical HRQoL [OR 1.58 (95% CI 1.01–2.46)], poor general health [OR 1.66 (95% CI 1.19–2.30)] and low overall quality of life [OR 1.54 (95% CI 1.21–1.96)]. However, this study focused on younger individuals (ages 20–39) without any pre-existing health conditions [45].

As for the correlation between napping and the improvement in mental components in people who have a short duration of sleep, this may be due to the effect that naps have on restoring cognitive functions, like increasing the level of alertness, activation, emotional improvement or vitality [25,27,46]. However, it is not yet clear what is the optimal duration of daytime sleep that allows such restoration. Some studies suggest that just 6 min of napping has restorative effects, with a 30 min nap duration yielding the strongest improvement [25]. On the other hand, long naps have been shown to be associated with increased mortality and morbidity with diseases such as depression, pain, obesity, type 2 diabetes or MetS [47,48,49,50]. Our results show beneficial effects for MCS, both in naps between 15 and 60 min/d and naps longer than 60 min/d with respect to sleeping less than 15 min/d, even if it is true that a stronger association is observed in individuals who sleep between 15 and 60 min/d. However, in the longitudinal analyses, we observed that a nap between 15 and 60 min/d is associated with a negative change in PCS in individuals with adequate or excessive night-time sleep.

Our study is not without limitations. The time elapsed between both measurements is only 1 year, which may be insufficient to detect clinically significant changes in HRQoL. Furthermore, our sample size was limited due to the lack of accelerometry data in some of the participants. In addition, it is necessary to recognize the limitations of accelerometry in that sleep duration is measured over a short period of time. Additionally, there is potential for selection bias, response bias, recall bias, unmeasured and unobserved confounding, time-varying confounding and misclassification of exposure. We must also highlight that our analyses have not taken the quality of sleep into account, which may be closely related to HRQoL [1,46]. However, our results are new and original, as there are no studies that evaluate the relationship between objectively measuring sleep duration at night and during the day and HRQoL in a large sample of elderly people with metabolic syndrome, both in a cross-sectional and in a longitudinal study over the course of one year.

## 5. Conclusions

In conclusion, the extremes in nocturnal sleep duration are related to lower PCS scores in the models adjusted for sociodemographic variables. Furthermore, a nap is associated with higher MCS scores in older adults who sleep less than 7 h a day but is postulated as a risk factor (or risk marker) for decreased HRQoL in people who sleep more than 7 h a night.

## Figures and Tables

**Figure 1 nutrients-16-02631-f001:**
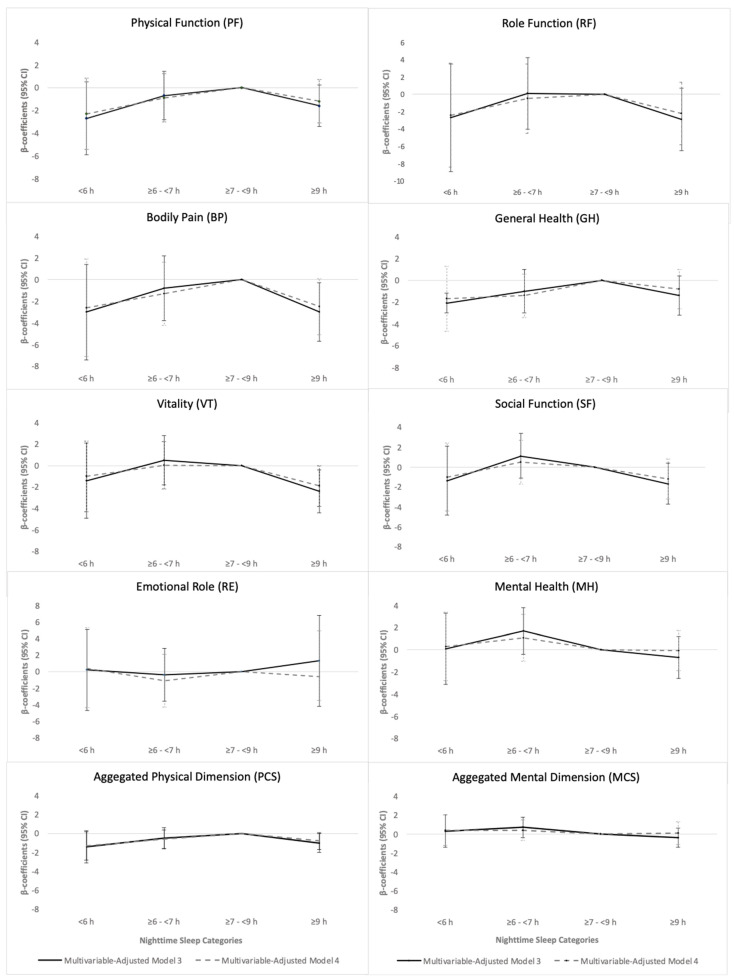
Multivariable-adjusted Model 3 and 4 β-coefficients (95% confidence interval) of dimensions of health-related quality of life according to night-time sleep duration categories.

**Figure 2 nutrients-16-02631-f002:**
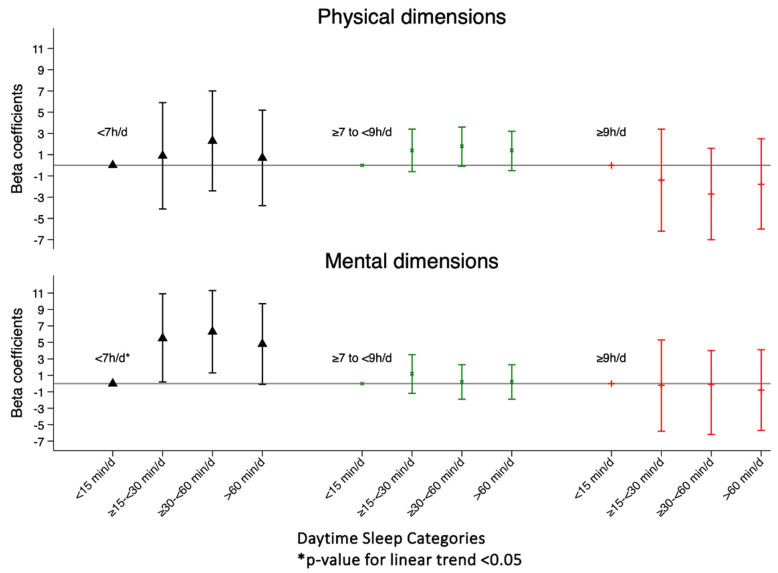
β-coefficients (95% confidence interval) in physical component summary and mental component summary according to daytime sleep duration categories, stratified by night-time sleep duration.

**Figure 3 nutrients-16-02631-f003:**
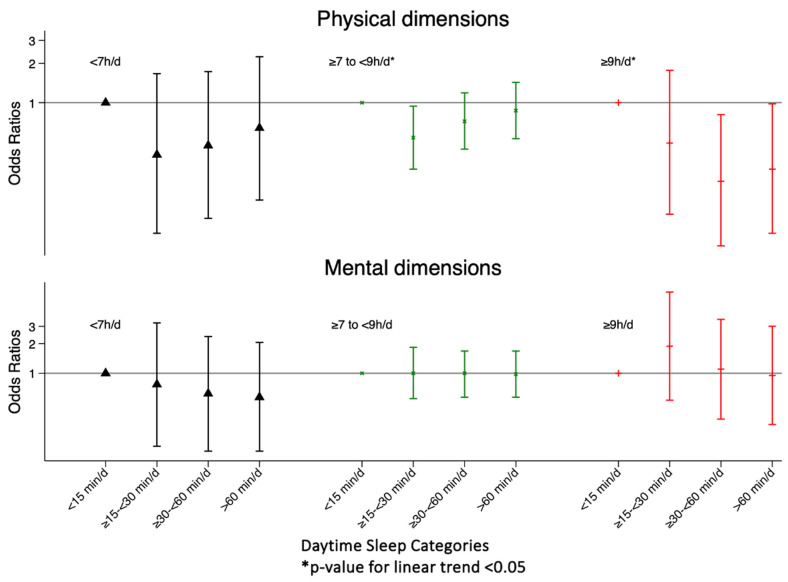
Odds ratios (95% confidence interval) of 12-month clinically significant changes in PCS and MCS according to daytime sleep duration categories, stratified by night-time sleep duration.

**Table 1 nutrients-16-02631-t001:** Baseline characteristics of the PREDIMED-Plus study participants across categories of night-time sleep duration.

	Categories of Night-Time Sleep Duration (h/d)					*p*-Value
	Total	<6	≥6–<7	≥7–<9	≥9	
	*n* = 2119	*n* = 129	*n* = 316	*n* = 1247	*n* = 427	
Sleep parameters						
Night-time sleep duration, Min–max, h/d	3.1–14.2	3.1–5.9	6.0–6.9	7.0–8.0	9.0–14.2	<0.001
Night-time sleep duration, mean (SD), h/d	8.0 (1.3)	5.2 (0.7)	6.6 (0.3)	8.0 (0.5)	9.8 (0.7)	<0.001
Napping duration, median (IQ), min/d	61.2 (37.8–91.2)	90 (69.0–139.8)	68.7 (42.0–103.2)	55.8 (33.0–81.6)	64.8 (40.8–97.2)	<0.001
Age, mean (SD), years	65.0 (4.9)	64.7 (5.3)	64.0 (5.1)	64.9 (4.9)	66.3 (4.4)	<0.001
Female, *n* (%)	1005 (47.4)	25 (19.4)	106 (33.5)	616 (33.5)	258 (60.4)	<0.001
Labor status, *n* (%)						
Retired	1219 (57.5)	77 (59.7)	162 (51.3)	698 (56.0)	282 (66.0)	<0.001
Educational level, *n* (%)						
≤Primary education	1062 (50.1)	56 (43.4)	129 (40.8)	611 (49.0)	66 (15.5)	<0.001
University education	463 (21.9)	40 (31.0)	95 (30.1)	262 (21.0)	266 (62.3)	
Smoking, *n* (%)						
Never	919 (43.4)	36 (27.9)	112 (35.4)	555 (44.5)	216 (50.6)	<0.001
Caffeine drinks/day, median (IQ), mg/day	21.4 (0–50)	21.4 (0–125)	7.1 (0–50)	21.4 (0–50)	3.3 (0–50)	<0.001
Alcohol drinks/day, median (IQ), g/day	5.1 (0.7–14.8)	7.4 (1.5–28.4)	7.3 (1.5–18.6)	5.1 (0.7–14.7)	2.9 (0.0–11.8)	<0.001
Leisure time spent watching TV, mean (SD), h/day						
Non-labor days	3.9 (3.3)	4.6 (7.7)	4.2 (5.2)	3.7 (2.0)	3.8 (1.9)	0.01
Sedative treatment, *n* (%)	514 (24.3)	29 (22.5)	52 (16.5)	295 (23.7)	138 (32.3)	<0.001
Depression, *n* (%)	472 (22.3)	23 (17.8)	51 (16.1)	274 (22.0)	124 (29.0)	<0.001
BMI, mean (SD), kg/m^2^	32.6 (3.5)	33.4 (3.5)	32.9 (3.4)	32.4 (3.4)	32.8 (3.6)	0.004
PA, mean (SD)						
IPA	7.2 (1.7)	8.9 (2.6)	7.6 (1.7)	7.1 (1.6)	6.6 (1.4)	<0.001
LPA	2.6 (1.1)	2.6 (1.3)	2.8 (1.1)	2.6 (1.1)	2.2 (0.9)	<0.001
MVPA	40.2 (32.2)	40.1 (34.6)	42.1 (33.4)	41.8 (32.5)	34.0 (28.9)	<0.001
HRQoL, SF-36 score, mean (SD), points (1-year follow-up)						
PF	79.3 (18.6)	78.3 (19.2)	80.9 (17.5)	80.2 (17.9)	75.7 (20.9)	<0.001
RF	81.3 (33.3)	82.2 (32.4)	84.2 (30.6)	82.1 (32.6)	76.6 (36.9)	0.01
BP	65.9 (25.1)	66.3 (24.6)	67.7 (23.6)	66.9 (24.9)	61.6 (26.5)	0.001
GH	64.3 (17.0)	63.8 (16.7)	64.9 (16.4)	64.9 (17.0)	62.0 (17.2)	0.02
VT	65.1 (19.3)	65.2 (17.6)	67.1 (17.7)	65.7 (19.1)	61.6 (20.9)	<0.001
SF	85.6 (19.3)	85.9 (20.4)	88.1 (17.5)	86.0 (19.2)	82.7 (20.0)	0.001
RE	90.3 (26.0)	92.2 (22.6)	91.1 (24.0)	90.6 (25.8)	88.2 (28.9)	0.27
MH	75.4 (17.8)	77.1 (16.0)	78.0 (15.5)	75.4 (17.7)	73.2 (19.6)	0.002
PCS	46.3 (8.4)	45.9 (8.5)	46.9 (8.0)	46.7 (8.3)	44.8 (9.1)	<0.001
MCS	51.5 (9.3)	52.3 (8.7)	52.4 (8.2)	51.4 (9.3)	50.6 (10.2)	0.05

BMI: body mass index; BP: bodily pain; GH: general health; h/d: hours/day; HRQoL: health-related quality of life; IPA: inactive physical activity (h/average day)—cut-off intensity level used for inactivity (excluding SIBs) < 40 mg (<1.5 Mets); LPA: light physical activity in bouts of at least 1 min (accumulated min/average day)—cut-off intensity level for LPA is >40 mg (1.5 Mets) and <100 mg (3 Mets); MVPA: moderate–vigorous physical activity in bouts of at least 1 min (accumulated min/day)—cut-off intensity level for MVPA is >100 mg (3 Mets); MH: mental health; MCS: aggregated mental dimensions; PA: physical activity; PCS: aggregated physical dimensions; RE: emotional role; RF: role function; SF: social function; VT: vitality. *p*-Value for differences between categories of night-time sleep duration was calculated by chi-squared test or ANOVA for categorical and continuous variables, respectively. In cases of non-normally distributed variables, we performed Kruskal–Wallis test.

**Table 2 nutrients-16-02631-t002:** Multivariable-adjusted β-coefficients (95% confidence interval) of health-related quality of life according to night-time sleep duration categories.

Categories of Night-Time Sleep Duration (h/d)				
	<6	≥6–<7	≥7–<9	≥9
	*n* = 129	*n* = 316	*n* = 1247	*n* = 427
HRQoL	β-Coefficients (95% CI) *p*-Value	β-Coefficients (95% CI) *p*-Value	β-Coefficients (95% CI) *p*-Value	β-Coefficients (95% CI) *p-*Value
PF	−5.4 (−8.6 to −2.3) 0.001	−1.7 (−3.9 to 0.5) 0.12	0 (ref.)	−2.2 (−4.1 to −0.2) 0.03
RF	−4.3 (−10.2 to 1.6) 0.16	−0.6 (−4.7 to 3.4) 0.77	0 (ref.)	−3.2 (−6.8 to 0.4) 0.08
BP	−5.0 (−9.4 to −0.6) 0.03	−1.8 (−4.8 to 1.2) 0.25	0 (ref.)	−3.4 (−6.1 to −0.7) 0.01
GH	−3.6 (−6.6 to −0.5) 0.02	−1.6 (−3.6 to 0.5) 0.14	0 (ref.)	−1.7 (−3.5 to 0.1) 0.07
VT	−4.1 (−7.5 to −0.7) 0.02	−0.5 (−2.8 to 1.8) 0.68	0 (ref.)	−2.9 (−5.9 to −0.9) 0.005
SF	−3.4 (−6.7 to 0.03) 0.05	0.4 (−1.9 to 2.7) 0.74	0 (ref.)	−2.1 (−4.2 to −0.08) 0.04
RE	−0.8 (−5.5 to 3.9) 0.73	−0.7 (−3.9 to 2.5) 0.66	0 (ref.)	−1.6 (−4.4 to 1.3) 0.29
MH	−1.5 (−4.6 to 1.6) 0.36	1.1 (−1.0 to 3.2) 0.30	0 (ref.)	−1.2 (−3.1 to 0.7) 0.23
PCS	−2.3 (−3.8 to −0.8) 0.002	−0.8 (−1.8 to 0.2) 0.10	0 (ref.)	−1.1 (−2.0 to −0.3) 0.01
MCS	−0.3 (−2.0 to 1.3) 0.70	0.5 (−0.6 to 1.6) 0.40	0 (ref.)	−0.6 (−1.6 to 0.4) 0.26

BP: bodily pain; CI: confidence interval; GH: general health; h/d: hours/day; HRQoL: health-related quality of life; MCS: aggregated mental dimensions; MH: mental health; PCS: aggregated physical dimensions; PF: physical function; RE: emotional role; RF: role function; SF: social function; VT: vitality. The data shown are those corresponding to Model 2 of the linear regression adjusted by sociodemographic variables: age, sex, marital status (married or living with a partner, divorced or widowed or single), labor status (active, retirement and unemployed or household work) and educational level (≤primary, secondary or university).

**Table 3 nutrients-16-02631-t003:** Multivariable-adjusted β-coefficients (95% confidence interval) of physical component summary and mental component summary according to daytime sleep duration categories, stratified by night-time sleep duration categories.

	Categories of Daytime Sleep Duration (min/d)						
	<15	≥15 to <30		≥30 to <60		≥60	
		β-Coefficients	*p*-Value	β-Coefficients	*p*-Value	β-Coefficients	*p*-Value
	Night-time Sleep Duration <7 h/d (*n* = 445)						
HRQoL, SF-36 Score							
PCS							
Model 3	0 (ref.)	0.9 (−4.1 to 5.9)	0.73	2.3 (−2.4 to 7.0)	0.34	0.70 (−3.8 to 5.2)	0.76
Model 4	0 (ref.)	0.9 (−4.1 to 5.9)	0.74	2.3 (−2.4 to 7.0)	0.34	0.98 (−3.6 to 5.5)	0.67
MCS							
Model 3	0 (ref.)	6.2 (0.8 to 11.6)	0.02	6.8 (1.8 to 11.9)	0.008	5.0 (0.1 to 9.9)	0.05
Model 4	0 (ref.)	5.5 (0.1 to 10.9)	0.04	6.3 (1.3 to 11.3)	0.01	4.8 (−0.1 to 9.7)	0.06
	Night-time Sleep Duration ≥7 to <9 h/d (*n* = 1247)						
HRQoL, SF-36 Score							
PCS							
Model 3	0 (ref.)	1.4 (−0.6 to 3.4)	0.18	1.8 (−0.1 to 3.6)	0.06	1.4 (−0.5 to 3.2)	0.15
Model 4	0 (ref.)	1.4 (−0.6 to 3.4)	0.17	1.9 (−0.1 to 3.7)	0.05	1.6 (−0.2 to 3.5)	0.08
MCS							
Model 3	0 (ref.)	1.0 (−1.4 to 3.4)	0.41	−0.1 (−2.3 to 2.1)	0.95	−0.5 (−2.6 to 1.7)	0.68
Model 4	0 (ref.)	1.2 (−1.2 to 3.5)	0.33	0.2 (−1.9 to 2.3)	0.87	0.2 (−1.9 to 2.3)	0.86
	Night-time Sleep Duration ≥9 h/d (*n* = 427)						
HRQoL, SF-36 Score							
PCS							
Model 3	0 (ref.)	−1.4 (−6.2 to 3.4)	0.56	−2.7 (−7.0 to 1.6)	0.41	−1.8 (−6.0 to 2.5)	0.41
Model 4	0 (ref.)	−0.7 (−5.4 to 5.4)	0.76	−2.2 (−6.5 to 2.0)	0.29	−0.7 (−4.9 to 3.4)	0.73
MCS							
Model 3	0 (ref.)	−0.5 (−6.2 to 5.2)	0.87	−1.1 (−6.2 to 4.0)	0.67	−1.6 (−6.7 to 3.4)	0.52
Model 4	0 (ref.)	−0.2 (−5.8 to 5.3)	0.94	−0.1 (−5.0 to 4.9)	0.98	−0.7 (−5.6 to 4.2)	0.77

HRQoL: health-related quality of life. PCS: aggregated physical dimension. MCS: aggregated mental dimension. The data shown are those corresponding to Model 3 and 4 because there was no change between Models 1, 2 and 3. Model 3: linear model adjusted by sociodemographic variables, such as age, sex, marital status (married or living with a partner, divorced or widowed or single), labor status (active, retirement and unemployed or household work) and educational level (≤primary, secondary or university), and lifestyle factors, such as smoking status (current, former or never smoker), caffeine drinks/day (mg/d), alcohol drinks/day (g/d), leisure time spent watching TV (h/wk), adherence to a Mediterranean diet assessed by a 17-item questionnaire (continuous), BMI (kg/m^2^) and MVPA recommendations (active/inactive). Model 4: linear model adjusted as in Model 3 and for morbidities, such as hypertension (yes/no), type 2 diabetes mellitus (yes/no), sedative treatment (yes/no), depression (yes/no), sleep apnea (yes/no) and chronic obstructive pulmonary disease (yes/no), and treatment assignment, stratified by night-time sleep duration.

**Table 4 nutrients-16-02631-t004:** Multivariable-adjusted odds ratios (95% confidence interval) of 12-month clinically significative changes in PCS and MCS according to daytime sleep duration categories, stratified by night-time sleep duration.

	Categories of Daytime Sleep Duration (min/d)						
	<15	≥15 to <30		≥30 to <60		≥60	
		OR (95% CI)	*p*-Value	OR (95% CI)	*p*-Value	OR (95% CI)	*p*-Value
	Night-time Sleep Duration <7 h/d (*n* = 445)						
HRQoL, SF-36 Score							
PCS							
Model 3	1.00 (ref.)	0.47 (0.11–1.92)	0.29	0.51 (0.14–1.90)	0.32	0.68 (0.19–2.44)	0.56
Model 4	1.00 (ref.)	0.43 (0.10–1.80)	0.25	0.50 (0.13–1.88)	0.31	0.73 (0.20–2.62)	0.63
MCS							
Model 3	1.00 (ref.)	0.66 (0.16–2.68)	0.56	0.64 (0.17–2.33)	0.50	0.59 (0.17–2.07)	0.41
Model 4	1.00 (ref.)	0.75 (0.18–3.19)	0.70	0.60 (0.16–2.30)	0.46	0.54 (0.15–1.95)	0.34
	Night-time Sleep Duration ≥7 to <9 h/d (*n* = 1247)						
HRQoL, SF-36 Score							
PCS							
Model 3	1.00 (ref.)	0.54 (0.31–0.93)	0.03	0.73 (0.44–1.19)	0.21	0.88 (0.54–1.44)	0.62
Model 4	1.00 (ref.)	0.53 (0.31–0.93)	0.03	0.72 (0.43–1.18)	0.20	0.87 (0.53–1.43)	0.59
MCS							
Model 3	1.00 (ref.)	0.98 (0.54–1.77)	0.94	0.97 (0.57–1.67)	0.92	0.99 (0.58–1.68)	0.96
Model 4	1.00 (ref.)	1.01 (0.55–1.84)	0.99	1.00 (0.58–1.68)	0.93	0.98 (0.57–1.68)	0.93
	Night-time Sleep Duration ≥9 h/d (*n* = 427)						
HRQoL, SF-36 Score							
PCS							
Model 3	1.00 (ref.)	0.53 (0.15–1.84)	0.32	0.28 (0.09–0.87)	0.03	0.33 (0.11–1.03)	0.06
Model 4	1.00 (ref.)	0.52 (0.14–1.89)	0.32	0.27 (0.08–0.88)	0.03	0.36 (0.11–1.15)	0.08
MCS							
Model 3	1.00 (ref.)	1.80 (0.51–6.36)	0.36	1.11 (0.35–3.49)	0.86	0.97 (0.31–3.02)	0.96
Model 4	1.00 (ref.)	1.87 (0.52–6.70)	0.34	1.09 (0.34–3.51)	0.88	0.94 (0.29–2.97)	0.91

HRQoL: health-related quality of life. Model 1: linear model adjusted for age and sex. Model 2: linear model adjusted as in Model 1 and for marital status (married or living with a partner, divorced or widowed or single), labor status (active, retirement and unemployed or household work) and educational level (≤primary, secondary and university). Model 3: linear model adjusted as in Model 2 and for lifestyle factors, like smoking status (current, former or never smoker), caffeine drinks/day (mg/d), alcohol drinks/day (g/d), leisure time spent watching TV (h/wk), adherence to a Mediterranean diet assessed using a 17-item questionnaire (continuous), BMI (kg/m^2^) and MVPA recommendations (active/inactive). Model 4: linear model adjusted as in Model 3 and for morbidities, such as hypertension (yes/no), type 2 diabetes mellitus (yes/no), sedative treatment (yes/no), depression (yes/no), sleep apnea (yes/no), chronic obstructive pulmonary disease (yes/no), daytime sleep (min/day) and treatment assignment, stratified by daytime sleep duration. Improved: increment 1 h night-time sleep duration across one year. Decreased: decreased 1 h night-time sleep duration across one year. Changes in HRQoL measured as improvement of 5 points in one year.

## Data Availability

The original contributions presented in the study are included in the article/Appendix A; further inquiries can be directed to the corresponding authors.

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
