# Peer review of "Objectively Measured Sleep Duration and Health-Related Quality of Life in Older Adults with Metabolic Syndrome: A One-Year Longitudinal Analysis of the PREDIMED-Plus Cohort"

_nutrients, 2024, doi:10.3390/nu16162631_

Round 1

Reviewer 1 Report

Comments and Suggestions for Authors

The authors evaluated daytime and nighttime sleep duration and health-related quality of life (HRQoL) in adults with metabolic syndrome after a 1-year healthy lifestyle intervention; by conducting cross-sectional and longitudinal secondary data analyses of data from the PREDIMED-Plus trial. They reported some positive findings in cross-sectional analyses but did not find statistically significant robust evidence of associations in longitudinal analyses. Sleep is a very important topic for public health. I thank the authors for their effort in conducting thorough analyses and for presenting an overall well-written manuscript. Suggestions:

1) In the abstract, please do not use terms like "increase", "decrease", or "improve" which would imply causal effects. Temporality is a big issue with cross-sectional design- so please use language appropriate for observational study- i.e. instead of saying "A increases B", one should say "greater prevalence of A is associated with higher B scores." ... Please edit the entire abstract accordingly, and similarly, change the conclusions of the abstract (lines 93-95).

2) Out of 6874, only 2,223 participants had accelerometer data. Please provide reasons for selection of subsample (was this intentionally selected subsample or was there very high rate of selective participation? if subsample was intentionally selected from larger sample then please provide selection criteria. One could also consider techniques such as inverse probability weighting as sensitivity analyses to address potential selection bias).

3) Were models adjusted for treatment assignment (i.e. assignment to control vs treatment arm to address potential confounding from potential Hawthorne effect)?

4) Was Bonferroni or other correction done for multiple comparisons testing? Please mention about it.

5) Please comment on minimal clinically important differences (MCIDs) in relation to the effect sizes for HRQoL in the discussion. For more information about MCID, one could refer to the following source: https://www.ncbi.nlm.nih.gov/books/NBK533976/

6) Please discuss and cite results from NHANES: Chen X, Gelaye B, Williams MA. Sleep characteristics and health-related quality of life among a national sample of American young adults: assessment of possible health disparities. Qual Life Res. 2014 Mar;23(2):613-25. doi: 10.1007/s11136-013-0475-9. Epub 2013 Jul 17. PMID: 23860850; PMCID: PMC4015621.

7) In the limitations para, please mention potential for selection bias, response bias, recall bias, unmeasured and unobserved confounding, time-varying confounding, misclassification of exposure.

Thank you for your important contributions.

Comments on the Quality of English Language

n/a

Reviewer 2 Report

Comments and Suggestions for Authors

Using PREDIMED-Plus data, the authors proposed a year-long longitudinal cohort and cross sectional study to evaluate older persons with metabolic syndrome's sleep quality and length of sleep.  The analysis was conducted on data from 2119 subjects aged 55-75 years. Findings suggested the relationship: reduced PCS is associated with nocturnal sleep duration extremes and napping increases the MCS for older adults who sleep fewer than seven hours per day.  

The study is very interesting and well conducted. Methods are clear and the paper is well written. 

Authors should consider the role of this association with cognitive impairment. A meta-analysis found that people with average sleep duration had a greater prevalence of metabolic syndrome, suggesting a strong correlation between sleep patterns and the development of metabolic syndrome and cognitive impairment (10.1002/alz.080568). Moreover, authors should consider the impact of frailty. Research indicates that lower heart rate variability and poor sleep quality are both independently and jointly linked to metabolic syndrome (10.1093/sleep/zsad013); this is relevant in frail population due to relationship (10.1186/s12877-021-02304-9).
